# A Review of Potential Feed Additives Intended for Carbon Footprint Reduction through Methane Abatement in Dairy Cattle

**DOI:** 10.3390/ani14040568

**Published:** 2024-02-08

**Authors:** Ian Hodge, Patrick Quille, Shane O’Connell

**Affiliations:** 1Department of Biological and Pharmaceutical Science, Munster Technological University, V92 HD4V Tralee, Kerry, Ireland; patrick.quille@mtu.ie (P.Q.); shane.oconnell@marigot.ie (S.O.); 2Research and Development Biotechnology Centre, Marigot Ltd., Shanbally, P43 E409 Ringaskiddy, Cork, Ireland

**Keywords:** rumen fermentation, enteric methane, mitigation, meta-analysis, feed supplementation

## Abstract

**Simple Summary:**

Introducing feed additives to mitigate enteric methane from ruminants demonstrates potential for reduced agricultural greenhouse gas emissions and opportunity for improved ruminant productivity. This review investigates garlic oil (GO), nitrate, *Ascophyllum nodosum* (AN), *Asparagopsis* (ASP), *Lactobacillus plantarum* (LAB), chitosan (CHI), essential oils (EOs) and 3-nitrooxypropanol (3-NOP) feed additives for methane (CH_4_) mitigation in large ruminants that have been investigated in in vitro or in vivo trials with the aim of improved rumen fermentation characteristics. Optimum dose ranges were determined from the literature and studies for each feed additive and were compared via meta-analysis. Feed additives were grouped based on in vitro or in vivo available studies, and conclusions were determined based on their effectiveness in live subjects or their potential efficacy in live animal trials. Standard mean differences of feed additives compared to the relative controls on both individual and summarised levels were used to determine rumen feed additive potential. 3-Nitrooxypropanal resulted in the greatest methane mitigating efficacy in vivo compared to nitrate and essential oil blends supported by promising VFA ratios and increased presence of hydrogen in favour of reduced enteric methane output. Furthermore, garlic oil, chitosan, and *Lactobacillus plantarum* displayed the potential for promising rumen fermentation alterations at their investigated in vitro levels. The active ingredient in Asparagopsis red seaweed, bromoform, elicits a more pronounced, dose-dependent methane mitigation effect compared to the primary compound found in brown seaweed *Ascophyllum nodosum*.

**Abstract:**

Eight rumen additives were chosen for an enteric methane-mitigating comparison study including garlic oil (GO), nitrate, *Ascophyllum nodosum* (AN), *Asparagopsis* (ASP), *Lactobacillus plantarum* (LAB), chitosan (CHI), essential oils (EOs) and 3-nitrooxypropanol (3-NOP). Dose-dependent analysis was carried out on selected feed additives using a meta-analysis approach to determine effectiveness in live subjects or potential efficacy in live animal trials with particular attention given to enteric gas, volatile fatty acid concentrations, and rumen microbial counts. All meta-analysis involving additives GO, nitrates, LAB, CHI, EOs, and 3-NOP revealed a reduction in methane production, while individual studies for AN and ASP displayed ruminal bacterial community improvement and a reduction in enteric CH_4_. Rumen protozoal depression was observed with GO and AN supplementation as well as an increase in propionate production with GO, LAB, ASP, CHI, and 3-NOP rumen fluid inoculation. GO, AN, ASP, and LAB demonstrated mechanisms in vitro as feed additives to improve rumen function and act as enteric methane mitigators. Enzyme inhibitor 3-NOP displays the greatest in vivo CH_4_ mitigating capabilities compared to essential oil commercial products. Furthermore, this meta-analysis study revealed that in vitro studies in general displayed a greater level of methane mitigation with these compounds than was seen in vivo, emphasising the importance of in vivo trials for final verification of use. While in vitro gas production systems predict in vivo methane production and fermentation trends with reasonable accuracy, it is necessary to confirm feed additive rumen influence in vivo before practical application.

## 1. Introduction

The three major greenhouse gases (GHGs) from the livestock sector are carbon dioxide (CO_2_), methane (CH_4_), and nitrous oxide (N_2_O). These GHGs contribute to climate change and the agricultural sectors carbon footprint through their absorption of infrared radiation in the atmosphere. CH_4_ and N_2_O are present primarily at two to six orders of magnitude lower than CO_2_ but absorb infrared radiation much more readily than carbon dioxide [1]. CH_4_ holds global warning potential (GWP) 25 times higher than that of CO_2_, making it a national and global concern in relation to atmospheric buildup [2]. A United Nations (UN) carbon offsetting initiative aimed at GHG-emitting industries was created to compensate for their unavoidable emissions by supporting worthy projects that reduce emissions elsewhere [3]. These carbon offsets are measured in UN Certified Emission Reductions (CERs) which provide the equivalent of one metric tonne of CO_2_ equivalents (CO_2_e) reduced or avoided. This initiative provides relevant industries with an opportunity to balance out their carbon footprints while reducing expenditure on carbon tax credits issued for emission of one metric tonne of CO_2_e. In agreement with carbon offsets, this study is designed to contribute to a strategy devised in October 2020 by the European Green Deal to combat CH_4_ emissions through multiple sectors within the EU and across the globe [4]. Globally, enteric fermentation emissions from cattle for beef and dairy account for 44 percent of GHG emissions from agricultural sources [5]. Methods of reducing the impact of biogenic CH_4_ to zero and carbon footprint reduction through methane abatement from the agricultural sector, as stated by the European Commission, are the aim of this study; we accomplished these aims through highlighting and collating data on CH_4_-mitigating feed additives for ruminants. The dietary additives discussed in this review were selected to explore the diverse mechanisms of CH_4_ mitigation in the rumen, including selective targeting of rumen methanogens [6], introduction of methyl-coenzyme M analogues [7,8], inhibition of protozoa [9,10], and shifting fermentation pathways, to promote alternative hydrogen sinks [11]. This review provides a comparison of commercial rumen products with different modes of action for enteric CH_4_ mitigation and highlights the presence of promising compounds for consideration in in vivo research. Additionally, this study incorporates additional targets for identifying feed additives capable of improving herd health status. The current end goal is the development of a modified feed for beef and dairy herds with the ability to increase productivity and reduce emissions per unit of product.

Enteric methane (CH_4_) is produced by a group of *Archaea* bacteria known as methanogens, which are commonly found in the rumen and hind gut of ruminant animals. These CH_4_-producing microbes belong to the Euryarchaeotic phylum and have the ability to produce CH_4_ liberally through the process of feed digestion [12]. Ruminant diets consist of plant tissue dry matter, which is about 75% carbohydrate, and contribute to the most important end products of carbohydrate fermentation. Volatile fatty acids such as acetate, propionate, and butyrate are produced as a result and are absorbed from the rumen as a major source of energy (70%) in the cows’ diet [13]. The breakdown of carbohydrates in the rumen can be separated into two stages; the first stage is the hydrolysis of complex carbohydrates (cellulose, hemicellulose, and pectin) into glucose equivalents by primary fermenters. The second stage involves the microbial degradation of these simple sugars into the main end products of rumen fermentation known as volatile fatty acids (VFAs), hydrogen (H_2_), carbon dioxide (CO_2_), and CH_4_ [14,15]. CH_4_ produced as a by-product of rumen digestive processes is identified as enteric CH_4_ and acts as the focus of reduction in terms of mitigation strategies for this review.

Oxidation of hydrogen (H_2_) using carbon dioxide (CO_2_) as an external electron acceptor is the favourable pathway in which CH_4_ is produced within the rumen [16,17]. Hydrogenotrophic H_2_ scavenging methanogens interact with other present ruminal microbes, including protozoa [18], bacteria [19], and fungi [20], which result in interspecies H_2_ transfer, thus producing CH_4_. This is also known as the hydrogenotrophic pathway, as illustrated in Figure 1 and adapted from Karekar et al. [21]. The methylotrophic pathway is an alternative CH_4_-producing pathway that utilises methyl groups, which can be sourced from methylamines and methanol substrates also described in Figure 1 [22]. Inter-microbial interactions occurring in the rumen benefit fermentation as they prevent H_2_ accumulation and feedback inhibition. Accumulation of H_2_ in the rumen can result in further re-oxidation of cofactors (NADH, NADPH, and FADH) and consequentially inhibit the production of volatile fatty acids (VFAs) [23]. The reduction of VFA production and absorption into the blood consequently limits the available VFAs as sources and precursors of energy, glucose, and non-essential amino acids in the liver [24]. Reduction of CO_2_ to CH_4_ acts as the largest H_2_ sink present in the ruminal pathways due to CH_4_ having the lowest (−2) possible oxidation state per unit of carbon [23]. By incorporating natural feed additives into the ruminant diet, a stress-free and direct approach to reducing enteric CH_4_ production can be achieved.

The forage to concentrate ratio can influence the fermentation rate in the rumen, causing an increase or decrease in enteric CH_4_ based on factors such as feed digestibility and passage rate [25]. Feeding with high-quality forages for the improvement of digestibility based on stage of growth at harvest and species was recently reviewed and tended to decrease livestock greenhouse gas contributions and improve productive efficiency [26]. Moreover, previous studies have demonstrated that the ratio of forage–concentrates has a notable impact on enteric CH_4_ emissions in cattle across various life stages [27], emphasising the need for suitable dietary strategies to accompany feed additives in ruminant systems. Varied forage quality and type have previously exhibited different amounts of easily fermentable carbohydrates available and passage rates through the ruminant digestive system, respectively [28,29]. An increase in easily digestible carbohydrates will lead to higher digestibility, which may also be accelerated by reduced microbial degradation required post fine chopping or pelleting of forage [28]. The diet fed to ruminant animals, especially the types of carbohydrates, are important in CH_4_ production as they possess the ability to alter the ruminal pH and subsequently modify the microbiota present [30]. The process of rumen microbial alteration as an intervention strategy for CH_4_ abatement requires special consideration as many factors such as feed conversion and animal productivity can be negatively affected. The mentioned feed additives in this study will be investigated through in vivo animal productivity and feed conversion results as well as in vitro VFA and CH_4_ production, based on availability from the relevant international published literature. Recent reviews have determined the ability of in vitro CH_4_ measurement to be indicative of trends in in vivo CH_4_ production; however, investigation of CH_4_ production relative to forage to concentrate ratios [31] and expression as a unit of feed digested [32] have previously diverged from simultaneous in vivo CH_4_ levels.

Significant interest has arisen in assessing alternatives for manipulating the microflora in the rumen of livestock due to the onset of public and regulatory demand for Europe to drop greenhouse gas emissions by 55 percent by 2030 compared to 1990 levels and become the first climate-neutral continent by 2050 [4]. Apart from the use of feed additives, reduction in CH_4_ production has been achieved using antibiotics with ionophore activity in the past [33]. Ionophores have been fed to cattle for decades to increase feed efficiency and act as antiporters that catalyse rapid ion movement across the cell membrane of rumen bacteria [34]. Inhibition of sensitive Gram-positive bacteria create shifts in rumen fermentation acids that are linked with enteric methane suppression [35]. Lack of social acceptance caused by residues found in food products and resistant strains of pathogens has forced the restriction of these products [36]. The restriction of antibiotics has led the focus on to natural compounds that possess the ability to mitigate CH_4_ by selectively targeting rumen methanogens, inhibit protozoa, stimulate propionate production, and act as alternative H_2_ sinks. The following rumen supplements will be investigated for enteric CH_4_ mitigation potential and maintenance of ruminant productivity in this meta-analysis review to determine their effectiveness in live subjects or their potential efficacy in live animal trials.

### 1.1. Garlic Oil

GO has been shown to exhibit antimicrobial activity against Gram-positive and Gram-negative bacteria [37]. By targeting cells directly in the rumen, the mechanism of action of garlic oil and its compounds are primarily related to direct inhibition of methanogenesis. Branched isoprenoid chains present in the *Archaea* lipid membrane described by De Rosa et al. [38] are susceptible to garlic through inhibition of the enzyme 3-hydroxy-3-methyl-glutaryl coenzyme A (HMG-CoA; see Figure 1) reductase, which catalyses the synthesis of the isoprenoid units found in the lipid *Archaea* membrane [39,40]. This mechanism serves as a threat to the methanogenic *Archaea* membrane and remains specific to this membrane’s lipid composition, relieving any threat to the remaining rumen microbiome’s unbranched fatty acid membranes. Some studies with GO supplementation have demonstrated inconsistent anti-methanogen effects in smaller ruminants with a trend towards increased methanogen population diversity [41]. The varied effectiveness of GO in mitigating CH_4_ emissions in the rumen may be linked to disparities in the quantities of bioactive compound found in the garlic or the composition of the basal diet to which the garlic supplement was added. However, increasing the dietary dose generally results in decreased CH_4_ production [42]. The natural compound’s potential for CH_4_ mitigation has been investigated extensively [9,43,44]; however, the exact active ingredient responsible for CH_4_ mitigation has not been determined. A mixture of plant secondary metabolites make up garlic oil, including allicin (C_6_H_10_S_2_O), diallyl sulphide (C_6_H_10_S), diallyl disulphide (C_6_H_10_S_2_), and allyl mercaptan (C_3_H_6_S) [45]. Although the secondary metabolites are similar in chemical structure, they have been studied individually in relation to rumen microbial fermentation by Busquet et al. [46]. Garlic oil diallyl and allyl compounds displayed similar but milder effects on CH_4_ mitigation and VFA production compared to garlic oil as a whole, suggesting a synergistic effect between the compounds present in garlic oil [46]. The toxic effect of the organosulphur compounds was avoided in this study by keeping both garlic oil and diallyl disulfide at a concentration of 300 mg/L of culture fluid to induce the observed CH_4_ reduction and a reduced acetate–propionate ratio. Inclusion of high levels of sulphur produces excessive rumen-generated amounts of sulphite, which may be absorbed in sufficient quantities to result in polio [47], a disease which causes blindness, staggering, and the inability to rise. The toxic effects can be avoided at low doses, but consequently, inhibition of methanogenesis diminishes also. The mitigation potential for garlic oil has been investigated within a range of 250–300 mg/L, an optimum range reported in the recent literature to be used in meta-analysis investigations.

### 1.2. Nitrate

Nitrate is an inorganic anion which acts as an alternative electron acceptor when introduced into the rumen [48]. Acting as an oxidising agent within the rumen, nitrate has a higher affinity for H_2_ than CO_2_, resulting in reduced enteric CH_4_ production [49]. Nitrate is biologically reduced to nitrite (NO_2_) when first introduced to the ruminant’s stomach, and it is then reduced again to produce ammonia. This pathway is energetically more favourable than the reduction of CO_2_ to CH_4_, limiting the hydrogen supply for the methanogen reduction reaction. Promising CH_4_ mitigation results have been observed through in vivo testing [10,48,50]. A drawback of nitrate’s H_2_ sink potential is the risk of nitrite build up in the rumen when doses of nitrate are introduced into the diet too quickly. An accumulation of nitrite can occur in the rumen and be absorbed into the bloodstream, resulting in the occurrence of methemoglobinemia [51]. This condition is caused by the oxidation of ferric iron in haemoglobin, thus rendering the red blood cells incapable of oxygen transport around the body. An in vivo study with sheep carried out by Alaboudi and Jones [52], as well as an in vitro trial with rumen fluid from steers [53], successfully cut the risk of methemoglobinemia through gradual introduction of nitrate into the diet, allowing time for microbe adaptation and increased nitrite reduction capability. Continuous loss in oxygen carriage can result in the severe poisoning of animals known as asphyxiation [52,54]. To avoid the risk of methemoglobinemia, nitrate was investigated at an optimum dose of 20–23 g/kg DMI and introduced into the rumen fluid gradually, as described [48,55]. Nitrate-metabolising bacteria that obtain energy for growth through reduction of such nitro compounds is another alternative to reducing the buildup of nitrite in the rumen. The abundance of *Campylobacter* and *Selenomonas* nitrate-reducing bacteria observed by Zhao et al. [56] increased prior to nitrate adaptation. Nitrate appears to be a capable candidate for CH_4_ mitigation as it has been reported to be effective and persistent over time in in vitro and in vivo trials.

### 1.3. Ascophyllum Nodosum

The brown algae *Ascophyllum nodosum* (AN) is found mainly along the coast in northern hemisphere waters, typically in sheltered rocky areas. Brown algae accumulate phlorotannins (PTs) as an adaptive technique to shield themselves from stress conditions and herbivory [57], setting them apart as the singular category of seaweed demonstrating this defensive mechanism. PTs are mainly found in brown seaweed in their free form or in the form of a complex within the cell wall of the plant. PTs are plant secondary metabolites in the form of an oligomer structure which consists of phloroglucinol units. The repeated 1,3,5-trihydroxybenzene units are linked with aryl-aryl, diaryl-ether or diaryl diether bonds [58]. Plant location appears to affect the PT content in brown seaweeds and seaweeds such as AN found in the intertidal zone, which contain higher levels of PTs compared to low-shore plants such as *Lamanaria digitata* [59]. AN has shown evidence of effective activity against rumen microbes when incubated with batch cultures, suggesting possible selective inhibition of microbes [60]. A possible explanation for this effect on the microbial population may be connected to AN’s ability to decrease protozoal activity [61]. A potential mode of inhibition of methanogens may be to target ciliate protozoa-dependent methanogens as a method of CH_4_ mitigation. Research carried out by Krumholz et al. [18] into the association of methanogens and rumen ciliates exposed the visual attachment of methanogens to the ciliate cell surface identified based on their specific fluorescence. Bright-field and epifluorescence produced fluorescent images of clusters and long chains of methanogens attached to the surfaces of both *Eudiplodinium maggii* (large ciliate) and *Eremoplastron bovis* (small ciliate), ciliates commonly found in the rumen. The pellicle of the ciliate exhibits the ability to enhance the growth and clustering of methanogens, thus acting as a target for enteric CH_4_ mitigation.

### 1.4. Asparagopsis

An additional macroalgae with promising enteric CH_4_-mitigating potential is a native Eastern Mediterranean red seaweed genus known as *Asparagopsis.* The species *Asparagopsis taxiformis* (AT) and *Asparagopsis armata* (AA) contain the active ingredient known as bromoform (BF) that has shown promise in the recent literature for its low inclusion levels and success in mitigating methane [11]. The growing body of research on bromoforms CH_4_-mitigating potential has prompted a recent review [62] emphasising the dose-dependent effectiveness of the active compounds across a range of ruminant systems. The halogenated CH_4_ analogue (HMA) compound is retained by the seaweed thanks to its distinctive cellular structure [63]. BF inhibits the final catalysis step in the methanogenesis pathway by removing the prosthetic group required by the enzyme methyl-coenzyme M reductase [64]. Variance in efficacy due to the difference in the BF content was demonstrated by Kinley et al. [11], who showed the greater CH_4_-mitigating potential of AT supplemented with 5.3 mg/g DMI additional BF content, compared to a study also carried out in cattle by Roque et al. [65]. Similarly, an incremental decrease in CH_4_ yield (g/kg DMI) associated with increasing bromoform supplementation is evident in in vivo studies feeding the different species, AA [11,66] and AT [67]. The life stage of the red seaweed was deemed a factor affecting the bromoform content by Paul et al. [63]. Wild harvesting of the species will also have an influence on the HMA’s BR, bromochloromethane, chloroform, and dichloromethane, as plants in different life stages will be picked from the sea floor during harvest. Bromochloromethane also displayed a methane reduction of around 30% while causing a decrease in the dominant methanogen *Methanobrevibacter* spp. population [68]. The cumulative effort of the HMAs stored in the vacuoles of the plant cells will be investigated both in vitro and in vivo in later sections.

### 1.5. Lactic Acid Bacteria

Lactic acid bacteria (LAB) are commonly used for their fermentation qualities, which improve the quality and digestibility of crop ensiling. LAB can be split into two groups that produce different end products during fermentation: homofermenters, which produce lactic acid as their product, and heterofermenters, which can produce multiple end products such as acetic acid, ethanol, and CO_2_. The buildup of organic acids lowers the pH of the silo contents, thus inhibiting the growth of pathogens and the threat of spoilage [69]. Forages ensiled with LAB are known for their ability for maintaining nutritional value and moisture content, but the basis of this process may be used to manipulate CH_4_ emissions.

CH_4_ emissions reduced by LAB have been reported [70,71,72], though special attention must be given to particular strains that possess the ability to mitigate enteric CH_4_ [73]. Different responses between studies are likely due to the strain, forage type, and method of ensiling. LAB have the ability to alter the propionic acid production rate in a silo by changing the amount of lactic acid and water-soluble carbohydrates present [74]. Increasing the amount of propionic acid being produced in the rumen has the potential to reduce CH_4_ production, as CH_4_ production is negatively correlated with propionic acid concentration [75]. *Lactobacillus Plantarum*, a homofermenting LAB, has demonstrated its ability to increase lactic acid after 45 days of ensiling compared with other LAB strains studied [76]. The *L. plantarum* strain also has a positive effect on propionic acid concentration while demonstrating CH_4_ output reduction in long-term in vitro studies [72,75]. In vivo studies have demonstrated improved digestion rates of *L. plantarum* using directly fed microbial methods; however, the *L. plantarum* strain is usually introduced with a combination of other LAB [77]. LAB-inoculated forages show significant reductions in CH_4_ due to increased propionic acid levels in vitro [70,72].

### 1.6. Chitosan

Chitosan (CHI) is the second most abundant natural biopolymer on earth after cellulose. CHI is obtained by the deacetylation of chitin, which is a fibrous substance consisting of polysaccharides and is usually found in the exoskeleton of arthropods and the cell walls of fungi. CHI possesses antimicrobial properties against bacteria, moulds, and fungi, which allow for the use of the biopolymer in a range of settings including food preservation and medicine [78]. The benefits associated with CHI activity in the rumen have been shown through altering propionate levels and methanogenesis [61,79,80]. Goiri et al. [81] found that the introduction of CHI can change ruminal fermentation by causing a shift in the VFA’s profile and increasing the propionate concentration; this shift in the fermentation of propionate regulates the amount of H_2_ available for the function of the methanogenic bacteria [82]. Varying the percentage of deacetylation of CHI can influence the impact the natural compound has on the rumen inhabitants. Goiri et al. [79,80] demonstrated in their research that the higher percentage deacetylation positively effects propionate concentration and thus CH_4_ emission in vitro; for this reason, CHI was found to be more effective compared to chitin (due to its degree of deacetylation). A medium to high dose of CHI with deacetylation above 85% displayed promising effects on VFAs and CH_4_; however, the in vitro digestibility is a factor which needs close attention, as it tends to decrease with increasing dose and percentage deacetylation.

### 1.7. Essential Oil Blends

Essential oil combinations have recently become commercially available as ruminant products with the aim of combating methane emissions and improving feed conversion in ruminants. The essential oil blends under investigation include a garlic and citrus extract (MO) (Mootral SA, Rolle, Switzerland), based on a proposed mode of action that involves garlic’s ability to inhibit HMG-CoA in methanogen membrane synthesis [40,83] and antimicrobial traits associated with flavonoids in citrus extracts [84]. Section 1.1. discusses garlic and its contribution to enteric CH_4_ mitigation leading to its probable consideration as a component of MO. Manipulation of rumen dietary pathways such as methanogenesis by garlic is reinforced by its influence on proportions of ruminal VFAs in line with reduced CH_4_ emissions [46,83]. The citrus extract flavonoid components have an inhibitory effect on methanogen populations [84], adding to the possible dual-pronged approach to CH_4_ inhibition by MO in the rumen environment.

The second essential oil commercial mix (AR) (Agolin SA, Bière, Switzerland) includes the plant extract active ingredients coriander (*Coriandrum sativum*), seed oil (10%), eugenol (7%), geranyl acetate (7%), and geraniol (6%), which make up this methane inhibitor; its mode of action in the rumen is still unclear. This plant-based feed additive was certified by Carbon Trust Assurance LTD for reduction of methane emissions and improvement of feed efficiency at a daily dose of one gram per cow [85]. The certified dose of 1 g/cow/day was used by in vivo studies included in this meta-analysis [86,87,88,89,90,91]. Possible modes of action for AR in the rumen include a shift in fermentation towards an increased propionate concentration at the expense of acetate in vitro [88]. Additionally, the active essential oils found in AR have previously been used in food for their antimicrobial properties, eugenol for treating beef and coriander for pork preservation [92,93]. Essential oils can interact with the cell membrane of Gram-positive and -negative microorganisms [94]. This method of inhibition can halt deamination and methanogenesis in the rumen, resulting in reduced ammonia, CH_4_, and acetate; in turn, it can increase propionate and butyrate production. The in vivo studies included in this review had similar CH_4_ data collection periods post AR supplementation [86,87,89], which allows for analysis of the impact of varied rumen adaption periods in each study.

### 1.8. 3-Nitrooxypropanol

3-nitrooxypropanol (3-NOP), commercially known as *Bovaer* (DSM), is a compound designed to inhibit methyl-coenzyme M reductase, the enzyme found towards the end of the CH_4_ pathway within the rumen. The inhibitor works by oxidising the enzyme and blocking the last step of methanogenesis [7], similar to the mechanism of CH_4_ mitigation observed with BF rumen supplementation. This direct inhibitory approach has been investigated through in vivo experiments that involve beef cattle [95] and dairy cattle [96,97,98,99]. From the available literature, 3-NOP has been used to treat cattle at a wide range of doses varying from 60 mg/kg DM [99,100,101] to 183 mg/kg DM [102]. The studies were compiled in two different groupings for meta-analysis, lower (60–75 mg/kg DM) and higher (100–183 mg/kg DM), to determine the optimum range for both CH_4_ mitigation and animal productivity.

## 2. Materials and Methods

A systematic literature review was constructed using peer-reviewed publications that involved the feed additive compounds of interest listed above. Publications were obtained from the Munster Technological University Scopus database [103] using key search words including CH_4_, rumen, and the natural feed additive of interest. No date range was established when accumulating publications regarding any of the search terms in Scopus, therefore eliminating this limiting factor. Criteria for each document to be integrated into the database required (1) the research to be carried out on CH_4_ emissions produced by beef and dairy cattle; (2) that CH_4_ emissions were directly measured using recognised methods and not through prediction methods; and (3) that investigations were carried out in vivo when adequate numbers were available, or were carried out in vitro where necessary. The listed supplementary feed compounds were selected based on their potential to be enteric methane mitigators and benefit animal productivity, and based on their signs of an optimum dosage rate.

A meta-analysis approach was used to compare the CH_4_-mitigating compounds of interest based on their mitigating potential relative to the average control values. Open meta-analysis software (version 5.26.14) was used in this meta-analysis-style review to identify compounds with significant CH_4_-mitigating efficacy while being safe for use in beef and dairy cattle [104]. The database for OpenMeta analysis was compiled using international peer-reviewed papers extracted from Scopus that involved the use of GO, nitrate, AN, ASP, *L. plantarum*, CHI, EO blends, and 3-NOP for enteric CH_4_ mitigation. Studies investigating the in vivo CH_4_ mitigation potential of GO, LAB, and CHI were limited, and therefore data from in vitro studies were included in the meta-analysis. The consistency of in vitro study inclusion for meta-analyses of GO, LAB, and CHI prevented their direct comparison to in vivo summaries due to the known limitations of in vitro trials. The in vitro rumen supplements display potential for enteric CH_4_ mitigation and rumen fermentation enhancement; therefore, they were included in this study for future in vivo consideration. There were also limited studies of AN and ASP for meta-analysis inclusion, so direct comparisons were completed between the available results.

Meta-analysis of data was conducted using standardised mean difference compared via an arbitrary unit and control groups in contrast to groups post inoculation of the investigated candidate after in vitro or in vivo incubation. The model statistics used were *p*-value, heterogeneity, and the point of estimate relative to the line of no effect. Multiple studies can be compared using this software (version 5.26.14) by generating a line of no effect displayed on a forest plot, in this case relative to CH_4_ mitigation, to decipher the positive or negative effects imposed on the in vitro or in vivo studies. In addition to CH_4_ emissions, other variables that were taken into consideration included gas production, VFA concentration, bacterial and protozoal counts, pH, digestibility, and ammonia concentration. Table 1 describes the search terms that were used to locate relevant papers for this literature review.

A total of thirty-seven papers from the papers meeting the selection criteria and optimum administration ranges were evaluated in meta-analyses using OpenMeta analysis software (version 5.26.14) for comparison in seven separate meta-analysis plots. The included studies were each divided into meta-analyses, as described in Table 2, Table 3, Table 4, Table 5, Table 6, Table 7 and Table 8. The preferred method of systematic review and meta-analysis (PRISMA) guidelines were used to identify studies (Figure 2, Appendix A) [105]. Briefly, selection of relevant in vivo as well as in vitro studies was based on the criteria mentioned in the previous section. Failure to meet this criterion resulted in the paper being discarded from the saved library of research papers. The optimum predicted concentrations investigated in each meta-analysis were GO (250–300 mg/L culture fluid), nitrate (20–23 g/kg DMI), *L. plantarum* (log 6–9 log CFU/mL), chitosan (16–50 mg/g DMI), essential oil blends (0.04–2.5 g/kg DMI), and 3-NOP at lower (60–75 mg/kg DMI) and higher (100–183 mg/kg DMI) dosages. Data extracted from papers were transformed into similar units of measurement for accurate comparison relative to dry matter intake (DMI), as varied species of ruminant have different adult weights which in turn would mean an increased DMI. Careful selection of optimum rumen fluid collection time, gas collection methods, and adaptation times will be based on the successful meta-analyses that we discuss.

## 3. Results

### 3.1. Garlic Oil

Figure 3 shows the published studies on GO that were included in the meta-analysis; they involve data generated using an in vitro experimental approach, a dosage range of 250–300 mg/L culture fluid, similar diets of mixed forage with concentrate, and large ruminant lactating animals, as listed in Table 2. The graphical representation details the statistical significance of garlic oil as a CH_4_ mitigation contender (Figure 3). The weight given to each study can be observed by the size of each box, which represents the influence each study has on the overall effect. The increased influence of Patra and Yu’s study due to retaining the least associated inverse variance can be observed when compared to the remaining studies [6]. The weight was distributed accordingly, as follows: [9] 15.20%, [6] 32.29%, [43] 30.01%, and [44] 22.49%. A closer examination of the plot reveals that the study of Soliva [9] is the only study with a confidence interval (CI) that does not cross the line of summarised effect, while also having the greatest point estimate out of the four studies. The remaining studies that also contribute to the summarised effect are statistically significant on an individual level, as devised by their positive standardised mean difference (SMD) values. The summary effect calculated at 4.963 SMD is identified by the dotted red line on the graph. The 95% CI generated a lower limit of 2.507 and an upper limit of 7.420. The weighted average is shown at the bottom of the plot as a diamond, with the centre point indicating the pooled effect size and its width displaying the pooled CI range. The pooled effect summary does not cross the line of no effect, represented as point 0,0 on the forest plot. The study’s heterogeneity is shown to be insignificant, as the effect sizes displayed alongside the forest plot include an I^2^ value of 56.555 and a probability value of 0.075.

### 3.2. Essential Oil Blends

Figure 4 displays the forest plot generated through a meta-analysis of essential oil blends MO and AR from the in vivo studies listed in Table 6. An estimated line of effect for the essential oil mixtures was generated from the included studies, with a value of 0.887. The CI values generated from the included studies generated a pooled summary that had a range of 0.312 to 1.416. The Carrazco et al. [90] study produced the lowest point estimates on the forest plot (Figure 4) and remained the only point estimate situated lower than the line of effect generated. Two of the AR studies’ CIs crossed the line of no effect in the generated forest plot [87,88], followed by a single MO study [106]. In terms of heterogeneity, the analytical results fell into the high-range category, producing an I^2^ value of 81.38%, which is considered significant due to its <0.001 *p*-value. An overall *p*-value of 0.002 for the meta-analysis model was calculated from the data inputs. The weight given to each study in the analysis was split based on sample size and generated the following percentages: 7.3% [86], 8.37% [87], 12.25% [88], 16.77% [89], 11.04% [107], 12.67% [90], 17.01% [106], and 14.54% [91].

### 3.3. Nitrate

The meta-analysis provided in Figure 5 demonstrates CH_4_ gas yield in the form of grams per kg of DM from nitrate treatment in ruminants. The forest plot was generated using similar units from the following in vivo studies [10,48,108,109]. These results, which were obtained following adjustment periods of the nitrate feed additive, allowed for adaptation and the avoidance of nitrite buildup, as mentioned in previous sections. Based on the box representation of weights on the plot, the weight appears to be quite evenly spread across each of the studies, with slightly more influence towards the study with a higher supplement dose [109]. The weight distribution for each study was split accordingly, as follows [109] 31.28%, [48] 28.07%, [10] 22.12%, and [108] 18.53%. Based on the I^2^ value (67.18%) and *p*-value (0.027), the heterogeneity of the overall meta-analysis can be deemed significant. The pooled effect size of each of these studies provided a summarised effect at 1.845 and an average 95% CI that extends from 0.732 to 2.958, which does not cross the line of no effect. On an individual level, studies carried out by van Wyngaard and Villar do cross the line of no effect [10,109]. The *p*-value given to this model at 0.001 demonstrates the statistical significance of this analysis.

### 3.4. Chitosan

The statistical analysis in Figure 6 was generated from in vitro trials mentioned in Table 5. These studies contributed to the estimated line of effect which had a value of 3.594, indicated by the dotted red line in the forest plot. The individual effect size of the three studies generated a pooled summary that ranged from 0.641 to 6.547 along the *x* axis. Heterogeneity was calculated and generated a I^2^ value of 81.9%, thus falling into the high heterogeneity range, with a significant *p*-value of 0.004. Plotted points generated by both studies by Belanche et al. overlap the study by Tong et al. in relation to Cis; however, the rumen simulation technique (Rusitec) investigation looks to be an outlier in this graphical representation [110]. The meta-analysis model was given a *p*-value of 0.017. The studies involved were given weight accordingly: 39.64% [61], 38.45% [111], and 21.91% [110].

### 3.5. 3-Nitrooxypropanol

Due to the divide in administered 3-NOP dosage, separate meta-analysis plots were generated from the lower and higher ranges in Figure 7 and Figure 8, respectively. The lower-dosage meta-analysis group contained eight published studies [95,96,99,100,101,102,112,113], while the higher-dosage grouping contained five studies [8,96,97,98,102]. Post meta-analysis, the calculated estimated line of effect produced values of 1.614 and 1.743 SMD for the lower- and higher-dose groups, respectively. The pooled summary produced a range of 0.801 to 2.427 in the lower-dose analysis range compared to the higher-dose analysis range of 1.064 to 2.244. The pooled effect summary represented by a diamond in both plots was observed to stay within positive values in Figure 7 and Figure 8. When *p*-values were generated, both higher- and lower-dose forest plot models produced significant *p*-values of <0.001. Weights given to the lower-dose studies include [101] at 12.47%, [100] at 7.85%, [95] 13.51%, [96] 13.89%, [99] 15.37%, [102] 12.44%, [113] 11.92%, and [112] 12.55%. In relation to the higher-dose analysis, weights of 13.37%, 16.53%, 17.51%, 21.51%, 15.26%, and 15.81% were given to studies by Haisan et al. [97], Reynolds et al. [98], Van Wesemael et al. [8], and Haisan et al. [96], as well as doses of 137 and 183 mg/kg DM by Melgar et al. [102], respectively. A significant heterogeneity value was detected for the lower-dose range of 3-NOP (<0.001) compared to the non-significant higher-dose plot *p*-value of 0.06.

### 3.6. Ascophyllum Nodosum

Research on supplementing ruminant diets with PTs derived from AN is a novel approach. Peer-reviewed papers on AN potential with similar inclusion rates are limited [114,115]. The lack of AN phlorotannin-related literature was insufficient to complete a meta-analysis on the topic, but strong evidence of protozoal depression [61] and linear reduction in vitro fermentation [60] demonstrate AN’s potential as a CH_4_-mitigating feed additive. Brown seaweed PTs were used in conjunction with polyethylene glycol (PEG), a tannin-complexing agent, to break already formed tannin–protein complexes that have the potential to decrease the utilisation of proteins and other nutrients in the rumen [116]. The effect of the PEG and PT complex was investigated by Wang et al. and resulted in a suppression of 17% enteric CH_4_ production over 24 h in comparison to no PEG [60]. This poses the following question: does the CH_4_ reduction in the presence of the PT and PEG complex outweigh the reduction in energy uptake due to the reduced deamination of amino acids in the rumen, and can this reduction be alleviated?

The AN phlorotannin CH_4_ reduction mechanism proposed by Belanche et al. [61] and based on a strong decline in protozoal activity was detected from doses as low as 1 g/kg. A linear correlation between protozoa concentration and gut bacteria degradation was also noted in the mentioned study, providing a need for the upkeep of the gut microflora population while reducing protozoal availability for methanogens. Similar to this 23% decrease in protozoal activity, evidence of protozoal decline was observed when condensed tannins from Leucaena were incubated with rumen fluid [117].

### 3.7. Asparagopsis

An anti-methanogenic effect corresponding with the BF content encapsulated in the specialised gland cells is demonstrated in vitro and in vivo at inclusion rates of *A. taxiformis* and *A. armata* from 0.5 g/kg DM to 6.4 g/kg DM intake [11,118]. Evaluation of VFA production uncovered no significant change in total VFAs; however, a decreasing trend in ratios of acetate to propionate was discovered in both in vitro and in vivo studies investigating AT [11,119]. Supplementation with 2 g/kg DM of AT containing 12 mg/kg DM bromoform demonstrated weight gain improvements with undetectable reduction in animal productivity in vivo [11]. Additionally, an in vivo study carried out by Roque et al. [67] investigated the minor inclusion of 5 g/kg DM AT with 39 mg/kg DM bromoform content, which induced a reduction in enteric CH_4_ yield by 79.8% supplemented with low-forage TMR and 52% for high-forage TMR. The mentioned inclusion rates of 0.2 and 0.5% *Asparagopsis* in ruminants were also reviewed by Eason and Fennessy [62], revealing promising reductions in CH_4_ emissions without any adverse health effects due to the lack of bioavailability in ruminants for the minimum effective bromoform dosage levels. A decreasing trend in DMI associated with 5 g/kg DM AT supplementation (39 mg/kg DM bromoform) based on a 14% decrease in steers [67] was matched by a similar decrease in feed intake of 10.8% with 4.6 g/kg DM AA in lactating cows [65]. A more recent study that investigated the effects of seaweed biomass extraction and the steeping of AA in edible vegetable oil to stabilise bromoform supplementation from the AT plant observed a 11% greater decrease in CH_4_ yield g/kg DM when the entire biomass of the seaweed was supplemented and not extracted by additional processing [118]. The varied procedures for preparing AA (through freeze-drying [65] and steeping the plant in vegetable oil [118]) induced almost identical CH_4_ mitigation with similar bromoform concentrations of 12.1 mg/kg DM and 16.9 mg/kg DM bromoform, respectively. The varying CH_4_ mitigation capacity of both AA and AT was accompanied by a trend of increased H_2_ production, indicating signs of hydrogen redirection away from CH_4_ formation [11,65,67]. A shortage in closely related Asparagopsis supplementation concentrations and related studies diminished the possibility of a meta-analysis.

### 3.8. Lactobacillus Plantarum

The meta-analysis observed in Figure 9 involves five studies [71,76,120,121,122] conducted to investigate the CH_4_ abatement effect of 6–9 log CFU/mL *L. plantarum* as a silage inoculant and directly fed microbial (DFM) supplement. The heterogeneity investigation generated an I^2^ score of 34.50; however, the related *p*-value of 0.191 indicated the result to be insignificant. The pooled summary of all study groups is positioned at a range of 0.046 to 1.526, and does not span across the line of no effect. The overall summery affect represented by the red dotted line indicates the result to be 0.786. On an individual study level, all studies except GUO in have lower CIs that cross the line of no effect. The statistical significance of this analysis was determined by a *p*-value of 0.037. The dataset is weighed out with a slight extra percentage of weight given to Monteiro [121] at 38.93% and O’Brien [122] at 24.21%, followed by Ellis [120] with 15.62%, Huyen [71] with 11.77%, and Guo [76] receiving 9.46% of the weight.

Furthermore, as well as reducing CH_4_ emissions, an increase in DM digestibility was a trend observed with studies investigating silage inoculated with *L. Plantarum* [71,76]. Contrary to this, an inconsistent effect on DM digestibility was observed, with DFM studies showing a reduction in digestibility and no significant effect in studies by Monteiro et al. [121] and Ellis et al. [120], respectively.

### 3.9. Comparison Meta-Analysis Plot

An effect on methane output can be observed in each of the individual meta-analyses mentioned above in Figure 3, Figure 4, Figure 5, Figure 6, Figure 7, Figure 8 and Figure 9. In line with directly comparing in vitro plots with in vitro and in vivo with in vivo plots, the pooled estimates of each supplementation candidate meta-analysis are included in the comparison in Figure 10 above. The labelled pooled estimates are stacked in order of greatest lower bound effect and are measured on a scale of SMD values. GO displays the greatest in vitro CH_4_-mitigating capacity in Figure 10, based on the compounds’ summarised effect of 4.95 and their CI values. The plotted summarised effect for CHI produced a competing in vitro SMD value (3.58) that was accompanied by the greatest variation observed. On the contrary, the LAB representation in Figure 10 shows the lowest in vitro estimate of 0.79, with greater precision based on CI representation. Nitrate, 3-NOP, and EO in vivo analyses were also compared, with both nitrate (1.85) and 3-NOP at low (1.61) and high (1.74) doses producing similarly sized SMD effect sizes and 95% CI ranges. The summary meta-analysis for EO had the lowest in vivo point estimate of 0.89, but did not show negative in vivo CH_4_-mitigating abilities (Figure 10). The variability of the in vitro studies involving GO and CHI emphasises the need for further validation in an in vivo system, using optimum dosage information gained from in vitro work.

## 4. Discussion

Several reviews have looked at CH_4_ mitigation options [62,123,124,125,126]. This study focused on CH_4_ emission intensity and reduction relative to animal production e.g., CH_4_ per live weight gain (LWG) and milk production (MP), while also taking factors such as dietary intake (DI), H_2_, CO_2_ output, and VFA production into account. Compounds such as GO, CHI, 3-NOP, nitrate, EO, and LAB were compared through a meta-analysis, while also considering seaweeds AN and ASP (which had insufficient studies for a meta-analysis approach).

The methods of CH_4_ mitigation mentioned above cover a wide spectrum of activity, ranging from direct methanogen inhibition, targeting ciliate protozoa, and the addition of alternative H_2_ sinks. Certain limitations must be considered when interpreting these results (and in vivo/in vitro methane abatement studies in general), including the potential for varied CH_4_ production in animals under the same feeding conditions [127]. The residual feed intake (RFI) of the animal can also vary from animal to animal, as lower RFI has been linked to lower CH_4_ emissions based on positive genetic correlation [128]. For this reason, controllable factors such as dosage, incubation parameters, and forage to concentrate ratio were used to create groupings for analysis.

Fifty percent of study groupings in this literature review were carried out in either in vitro or in vivo studies. As the nitrate, essential oil blends, and 3-NOP data were collected from trials carried out on living ruminant hosts which also included incubation periods, results cannot be compared directly against in vitro analyses within this review but can be used to determine their potential as products developed for use in farm environments. The generated effect size and pooled effect size from each forest plot form the main comparison between each of the groupings. Taking these provisos into account, garlic oil demonstrated the greatest potential for methane mitigation in vitro (Figure 3 and Figure 10). The study conducted by Soliva illustrated the greatest mitigation of 7.23 mmol CH_4_ per day (91%), relative to the mean value in the forest plot representation for garlic oil (Figure 3). The ratio of forage to concentrate fed in the isolated study by Soliva [9] shows a noticeable difference when compared to the high-concentrate diets of the remaining studies. Hay was fed ad libitum to brown Swiss cows in the aforementioned study [9], enabling the conversion efficiency of cellulose and hemicellulose to produce sugars, amino acids, and liberated molecular hydrogen [129]. The increased population of methanogens for the reduction of CO_2_ at this stage in ruminant digestion may have created an increased number of targets for garlic oils antimicrobial behaviour, thus suppressing the CH_4_ biproduct. Both the batch and continuous fermentation studies from this study demonstrated a decrease in the proportion of acetate to propionate. Patra and Yu conducted a study which had the next greatest CH_4_ mitigation (by 6 mL/g DM (29%) from the control value [44]) and included data collected from a 48 h in vitro study with GO supplemented at 0.05 g/L less than the values of Soliva et al. [9]. This in vitro study continued for a further 16 days, during which time a slight decrease in methane mitigation was observed. This shift towards a decline in CH_4_ mitigation could indicate an inclination towards heightened methanogen diversity or potential resistance to GO in long-term in vitro studies. A possible solution to this short-term impact might involve the combination of GO with other dietary strategies or a mixture of EOs to sustain CH_4_ mitigation over extended durations. The study by Soliva et al. being the only Rusitec method used in the meta-analysis could possibly be the reason for the greatest effect size and CI range [9]. The Rusitec in vitro system has been in use for over 40 years [130], continually evolving as a rumen microbial process simulator without the risk of variability associated with live subjects in animal trials. The standardised Rusitec environment allows for continuous rumen fermentation simulation with the aid of inlet and outlet ports for environmental regulation. Reduction in protozoal counts is a reoccurring theme with GO supplementation and its active components (allicin) [6,131], which may support the inclination towards the indirect suppressing effects of GO as a rumen supplement.

The greatest summary effect size for 3-NOP was produced by the higher 3-NOP dosage (Figure 8), suggesting the requirement for supplementation within a range of 100–183 mg/kg DMI of 3-NOP to induce sufficient CH_4_ mitigating results. However, significant reductions of 31.4% and 27.3% in CH_4_ yield (g/kg DMI) were observed by Hristov et al. [101] and Melgar et al. [112] in the low-dosage 3-NOP meta-analysis (Figure 7). The supplementation size of 3-NOP in the mentioned studies lay in the lower fraction of the low-dosage range at approximately 60 mg/kg DMI and displayed positive effects on the rumen emission ratio. Increases in hydrogen emissions post 3-NOP administration detected by greenfeed systems [99,100,101,102,112] and reported shifts in VFA concentrations towards propionate proportions [97,98,99,132] indicate the possible redirection of excess H_2_, as an intermediate in CH_4_ formation, within the rumen to other net sinks of hydrogen. Despite the redirection of H_2_ from methanogenesis in most 3-NOP studies, improved productivity signals were inconsistent with occasional improvements in milk components [101,102,112]. As the included studies largely involve lactating cows, further studies investigating beef cattle may uncover improved productivity through VFA profiles and average daily gain values associated with 3-NOP supplementation and its ability to redirect H_2_ to nutritionally beneficial rumen pathways. Large CI ranges produced by Hristov et al. [101] and Lopes et al. [100] in the low-dosage meta-analysis (Figure 7), with three study CI ranges crossing the line of no effect, indicate that the higher doses of 3-NOP are more successful and consistent in mitigating methane. The pooled effect estimates for both high and low 3-NOP dosages have similar positive values for SMD of 1.74 and 1.61, respectively. However, the higher 3-NOP studies’ CH_4_-mitigating effects appear to be more reproducible and consistent based on the point estimates plotted. The study conducted by Reynolds et al. [98] was an anomaly, being the only study with a CI that crossed the line of no effect (Figure 8). Reynolds included the highest dose of 3-NOP in terms of grams per day at 2.5 g/cow/day, which was dosed with morning and evening feeding directly into the rumen via fistula [98]. This method of 3-NOP administration differs from the other studies in Figure 8 and may have caused rapid absorption within the stomach or washed out with liquid outflow—a process described by Seo et al. [133]. A gradual inclusion by mixing with feed or controlled release using concentrate pellets for such a high dose of 3-NOP may act as a more beneficial delivery method. Allowing the ruminant systems to undertake periods of adaptation to the 3-NOP treatments for at least two weeks also appears to be beneficial in terms of CH_4_ mitigation [8,97]. The effectiveness of adaption periods has been seen previously for GO [44,46] and nitrate [48]. Sudden inoculation with nitrate feed supplements, especially at high doses, appears to shock the rumen digestive system and negatively affect the DMI of treated animals [50]. Studies included in the nitrate meta-analysis demonstrated the benefits of nitrate adaptation in the diet by inducing CH_4_-mitigating effects (Figure 5) while maintaining DMI levels.

The *L. plantarum* meta-analysis (Figure 9) plot revealed interesting effect sizes for the studies that applied the LAB treatments as silage inoculants [71,76]. The effect sizes illustrated in Figure 9 suggest over double the CH_4_-mitigating capacity of *L. plantarum* when used as a silage inoculant in vitro. Traditional methods for preventing silage from spoilage and preserving nutritional value appear to induce further rumen function benefits. Substantial ensiling periods of 45–60 days with 10^6^ colony-forming units per gram of fresh weight forage increased DM digestibility by 42.7 g/kg DM (7%) and CH_4_-mitigating capacity by 8.8 mL g^−1^ (11%) of the inoculated forage in studies by Huyen et al. and Guo et al., respectively [71,76]. On the contrary, studies investigating cows treated with *L. plantarum* as a DFM failed to produce a standardised mean difference above the value of 1.0. A plausible reason for the reduction in total gas production and VFA concentrations observed with *L*. *plantarum* DFM in short-term trials may be the occurrence of microbial disruption within the rumen. Direct introduction of a non-dominant genus such as *L. plantarum* into the rumen microbial population has the potential to reduce cross-feeding, and in this case, an inhibition of rumen fermentation and a drop in rumen productivity were evident [121,122].

The Rusitec method used in the outlying CHI study included a 10-day stabilisation period in the artificial rumen before an 8-day data collection period [110]; Figure 6). It is possible that the Rusitec design creates a more realistic simulation of adaptation in the rumen with added supplements, in comparison to 24 h analysis of CHI supplementation. The shorter in vitro experiments create the conditions of the initial adjustments made by the rumen fluid in relation to the added supplement, whereas Rusitec allows for the investigation of gradual changes over longer periods, avoiding animals’ variability in a standardised environment [134]. A lack of ruminal absorption processes from the rumen epithelial surface area may be a reason for the accumulation of propionate mentioned in the results [110]; however, propionate buildup was also observed in the short term in vitro studies [70,71,76]. Furthermore, the reduction Belanche et al. [110] observed in gas production and microbial counts suggest CHI caused a shift in the structure of the bacterial community. This may be due to electrostatic interactions between the multiple positive structural sights associated with CHI and the electronegative sites found on the peptidoglycan wall of rumen microbes ultimately causing cell lysis [135]. A shift of fibrolytic abundance to amylolytic has the potential to increase amylase activity and in turn increase propionate concentration. Another propionate-related result proposed by a different study [79] suggests a shift in the VFA profile that increases propionate concentration caused by CHI-correlated rumen fermentation.

The remaining 24 h in vitro studies in Figure 6 decrease in methane-mitigating effectiveness as the dose of supplemented CHI increases. The CH_4_ data summarised in Figure 6 do not pinpoint an effective CHI supplementation range, but it can be concluded that at least 50 mg/g DM of CHI has the potential to significantly mitigate CH_4_ when supplementing a Rusitec in vitro system; however, in vivo studies are required to verify that 50 mg/g DM CHI supports the anti-methanogenic properties shown in Rusitec with no further possible side effects on feed intake, palatability, and animal performance.

Nitrate analysis produced significant results in vivo when investigated at 20–23 g/kg DMI inclusion rates. Research carried out by Klop et al. [108] contradicted other studies, with an SMD value of 3.706 that was almost double that of the next study (Figure 5). The remaining studies are situated close to the summary effect size, with some studies’ CIs merging into the negative SMD values across the line of no effect. Point estimates in Figure 5 that lie beyond the summary effect [48,108] are from studies that included adaption periods that lasted three weeks or longer and in which 21 g/kg of DM was given. Adaption periods appear to be a successful strategy for preventing methemoglobinemia while also acting as an alternative electron acceptor [136], hence the reduction of 3.3 and 5.1 g/kg of DMI CH_4_ output, respectively [48,108]. The lack of rumen adaptation to nitrate was observed by Lund et al. in their 24 h in vivo study [50]. Nitrate applied to the animals through feed inoculated with nitrate induced a direct effect on feed intake in the first four hours after the beginning of the data collection. Slightly increasing nitrate supplementation for 3–4 weeks in ruminants allows for palatability-friendly feeding as well as the familiarisation of the rumen microbes with the novel feed additive. However, studies from van Wyngaard et al. and Villar et al. [10,109] both fell short of the summary effect line, yet both included a 3-week adaption period. It is possible that the increase in pasture DMI observed by van Wyngaard et al. [109] may have increased the passage rate and caused the outflow of the nitrate supplement. In agreement with Villar et al. [10], the reduced time the supplement suspectedly spent in the rumen could have resulted in incomplete nitrate reduction for CH_4_-mitigating effect. Despite this possibility, a significant decrease in CH_4_ production was observed in both studies, with evident results of pH stability (pH 6.4–7) seen with 23 g of nitrate per kg of DM [109].

The meta-analysis plot that represents AR and MO essential oil products produced results of variable CH_4_-mitigating efficacy (Figure 4). Most point estimates representing both essential oil blends have positive SMD values; however, due to large CIs for AR studies and the lack of in vivo studies for MO, identification of the most effective essential oil blend is challenging. Both AR studies carried out by Klop et al. were carried out over different experimental period lengths but produced similar results in terms of a transient mean decrease of 9.5% in CH_4_ production [87,88]. Klop et al. conducted a study over 22 days and observed a reduced CH_4_ mitigation trend after 8 days [88]. Similarly, Klop et al. conducted a study over a 10-week period and observed the same trend after the first 2-week period [87]. An explanation for this shift may be the adaptation of rumen microbes to the AR supplementation. As mentioned above, the initial effect of AR on CH_4_ production may account for the initial adjustment period within the rumen as the resilient microbes return to their normal counts despite the antimicrobial effects of the novel supplement. The effect of essential oils on ruminal protozoa and methanogen numbers discussed by Patra and Saxena [137] shines a light on AR’s potential to change microbial counts within the rumen environment. Similarly, essential oil antimicrobial activity may also be attributed to the loss of microbial diversity after the recommended dose of AR was supplemented for 13 weeks [91]. The results show that methanogenic microbes are being targeted by AR; however, it is possible that the inclusion of antimicrobial essential oil blends may also be inhibiting essential digestive microbes, resulting in the reduction of feed efficiency [91] and milk production [86]. A possible explanation for the increased DMI in AR studies would be that the trial cows are trying to compensate for reduced energy uptake from their diet due to the inhibition of beneficial feed-digesting rumen microbes [89,91]. Certain species of Holstein rumen microbes have previously displayed their ability to increase in concentration while other species remain sensitive [138].

The two included MO studies were carried out on crossbred steers and resulted in significant reductions in CH_4_ yield with little to no improvements in productivity parameters [106,107]. Trends in CH_4_ mitigation with the two essential oil blends differed, as significant inhibition of CH_4_ was not detected until week 12 of the MO trial [107], compared to the demise of CH_4_ mitigation after rumen adjustment to AR supplementation [87,88]. The included MO studies suggest the resilience of the garlic and citrus extracts’ CH_4_ mitigation capabilities to rumen adaptation, with evidence of significantly inhibited CH_4_ emissions after 122 days of MO supplementation [106] and gradual increases in CH_4_ mitigation over 12 weeks [107]. In a recently published trial not included in the forest plot, MO-supplemented cows endured CH_4_ mitigation with a garlic and citrus extract blend without redirection towards significant productivity in terms of rumen fermentation characteristics [139]. Further post-MO supplementation microbiota profiling of a higher abundance of bacteria with hydrogen-scavenging characteristics and lower abundance of CH_4_-producing archaea, supporting the findings of Khurana et al. [139], may identify MO’s mode of action in the rumen.

Based on previously discussed points, the decrease in protozoal numbers caused by AN suggests indirect methods of methane mitigation. Ciliate association with methanogens described by Krumholz et al. [18] suggests the long chain growth patterns of methanogens spawned from ciliate interaction. A direct decline in protozoa caused by the supplementation of AN with PEG has shown promising signs of decline in methanogen activity. The use of PEG as a method of determining rumen fluid volume [140] has since evolved into a method of reducing the adverse effects of tannins when introduced into the rumen [141]. Anti-nutritional effects caused by tannins can be alleviated by the formation of strong hydrogen bonds of oxygen molecules in PEG with the phenolic and hydroxyl groups in tannins [142]. The inclusion of PEG with PT demonstrated an increase in rumen productivity including VFAs and a decrease in the acetate to propionate ratio over 24 h, which in turn reduced the amount of CH_4_ produced [60]. Thus, supplementation with AN and PEG at 500 μg/mL and 1 mg/mL, respectively, has the potential to generate positive effects on both rumen functionality and methane output in vitro over 24 h. Further investigation over longer in vitro experimental periods and in vivo trials are required.

The difference in anti-methanogenic activity between the brown seaweed AN and the red seaweed Asparagopsis is their bioactive ingredient, bromoform, which acts as a halogenated methane analogue. HMAs act as the active methane inhibitors in AT in comparison to PTs found in AN. HMAs’ mechanism of action in the rumen differs to that of PTs by binding and occupying the prosthetic group required by the methyl-coenzyme M reductase in the process of CH_4_ production [64]. In vitro studies allowed for elevated inclusion rates of AT, as palatability and daily intake are not considered in the absence of live animals; however, Kinley et al.’s [119] study in particular confirmed VFAs’ stability over 72 h at a rate of 50 g/kg DM. Long-term supplementation of AT in vivo was verified at lower rates of 2 g/kg DMI for 90 days [11] and 5 g/kg DM for 147 days [67]. The extended AT exposure caused an extreme reduction in enteric CH_4_, reaching almost complete mitigation (98%) in the study compiled by Kinley et al. [11], at 2 g/kg DMI. No decrease in DMI was apparent after the 90-day study [11]; conversely, a decrease of 14% DMI at a similar dosage was detected after 147 days, which requires further investigation in vivo to identify any detrimental effects associated with decreasing DMI. The forage content variations of the TMR diets used by Roque et al. [67] displayed a significant decrease in DMI, with 5 g/kg DM AT (39 mg/kg DM bromoform) inclusion across all forage variations, while maintaining the average daily gain of the trial steers. This drop in DMI was accompanied by a sizable increase in CH_4_ production (40%) when feeding with the high-forage diet compared to the lower forage inclusion. This study suggests that AT supplementation with high-forage diets is capable of increasing feed efficiency in beef cattle while significantly reducing CH_4_ emissions. The DMI decrease may also have been diet-specific, as there was no effect on DMI with the red seaweed species supplemented with higher-grain diets [11,118], contrary to TMR diets [65,67]. Analysis of the content of the active ingredient of Asparagopsis, bromoform, reveals a dose-dependent response in terms of CH_4_ mitigation and a more accurate method of administration to cattle when determining the optimum vehicle for delivery to the rumen [62]. Further in vivo studies will be capable of determining the influence of diet content with Asparagopsis supplementation on DMI, with the requirement of further VFA profiling to identify the improved redirection of H_2_ from CH_4_ production.

The pooled estimate comparison (Figure 10) of each meta-analysis displays an overall clear methane mitigation effect. Based on this meta-analysis, the standout in vitro candidate was GO, followed by CHI, a supplement that requires further trials to confine CIs and determine its true CH_4_-mitigating potential in vivo. Similarly positioned pooled estimates were produced by nitrate and 3-NOP, closely followed by EOs when aligned. In correspondence with individual meta-analysis pooled estimates, none of the pooled estimates crossed the line of no effect in Figure 10, showing confidence in the methane mitigation potential of each of the remaining candidates. However, comparison studies unveiled the promising CH_4_ mitigation potential of in vitro candidates GO and CHI as directly fed additives; they require in vivo studies to compare them directly against commercial EO products and upcoming 3-NOP supplements. An inquiry into their methane reduction potential at each of the agreed doses in vivo is required following this review.

## 5. Conclusions

The meta-analysis results produced for each of the included feed additives display positive CH_4_-mitigating potential based on the positive pooled estimate values and individual point estimates. Initial adaptation periods of at least 3 weeks after the introduction of nitrate into diets are vital to allow for the rumen microbiome to adjust and avoid rapid absorption or washout. Similarly, LAB treatments are more productive in the rumen when gradually included as silage inoculants, as opposed to being directly fed microbial populations. Ciliate counts in correspondence with CH_4_ production will determine indirect CH_4_ inhibition from GO, EO, and AN supplementation. ASP has a strong influence on rumen fermentation, with particular attention required for microbiome population modifications post inclusion with PT concentrations. The determined CH_4_-mitigating efficacy of 3-NOP at both high and low doses proves its potential as an effective enteric CH_4_ mitigator that requires specific administration doses to be identified. Future research and meta-analyses may benefit from more published studies becoming available featuring specifically chosen optimum doses for certain feed additives. GO showed the greatest CH_4-_mitigating potential in vitro at 300 mg/L during batch trials; in vivo measures are required to identify the CH_4_-mitigating significance of various feed additives compared with that of CH_4_-mitigating essential oils and 3-NOP commercial products.

## Figures and Tables

**Figure 1 animals-14-00568-f001:**
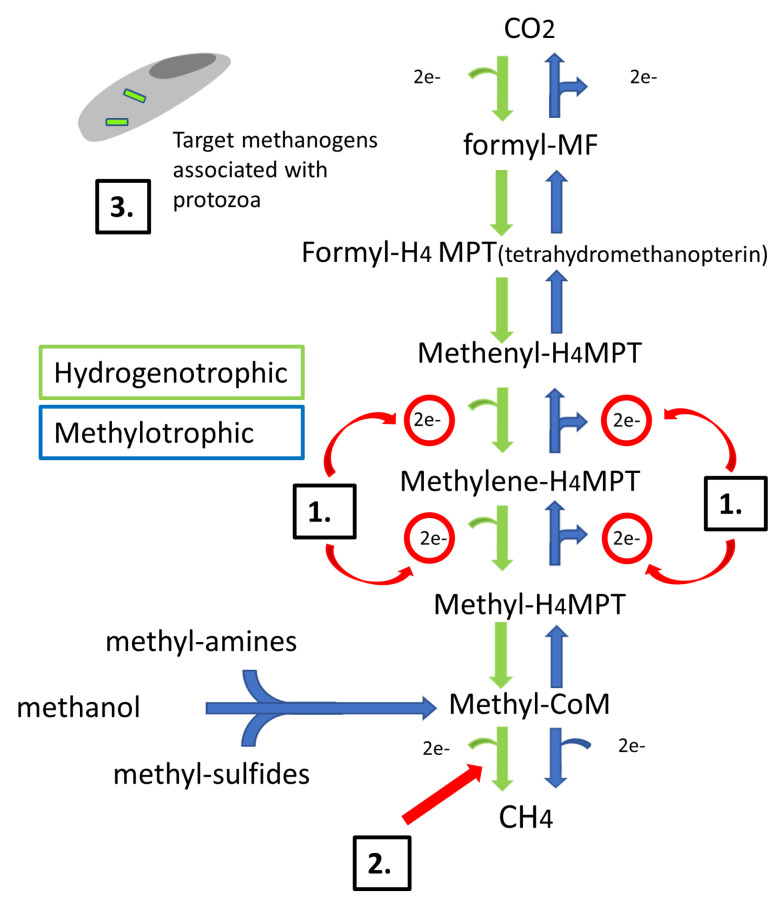
Hydrogenotrophic and methylotrophic pathways producing CH_4_ product from H_2_/CO_2_, methanol, methylamines, methyl sulphides as substrates for methanogenesis. Feed additive methods of CH_4_ mitigation include (**1.**) H_2_ scavenging of H_2_ oxidation to H^+^ in the hydrogenotrophic pathway and H_2_ accumulation from the methylotrophic pathway; (**2.**) inhibiting 3–hydroxy–3–methyl–glutaryl coenzyme A (HMG–CoA); and (**3.**) targeting membranes of ciliate protozoa hosting dependant methanogens. Figure adapted from Kracker et al. [21].

**Figure 2 animals-14-00568-f002:**
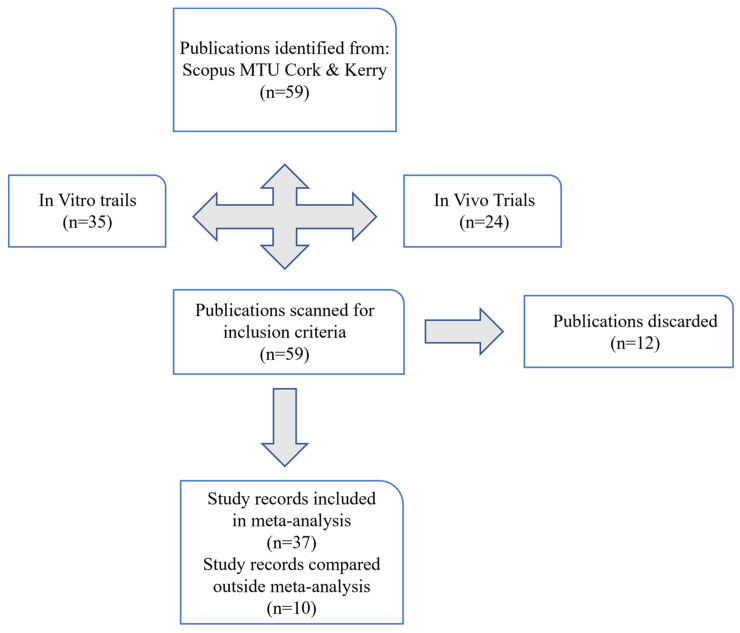
PRISMA flow diagram of published papers received from the Scopus search engine and included in the meta-analysis.

**Figure 3 animals-14-00568-f003:**
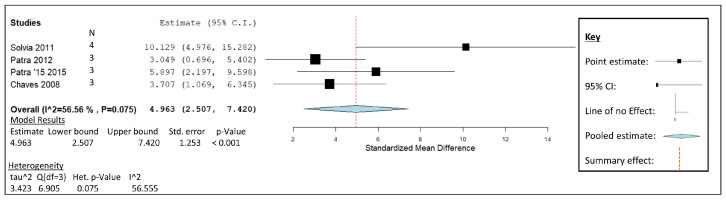
Forest plot representation of garlic oil (250–300 mg/L culture fluid) meta-analysis in vitro. C.I.—confidence interval; std. error—standard error; Het. *p*-value—heterogeneity *p*-value. References included [6,9,43,44].

**Figure 4 animals-14-00568-f004:**
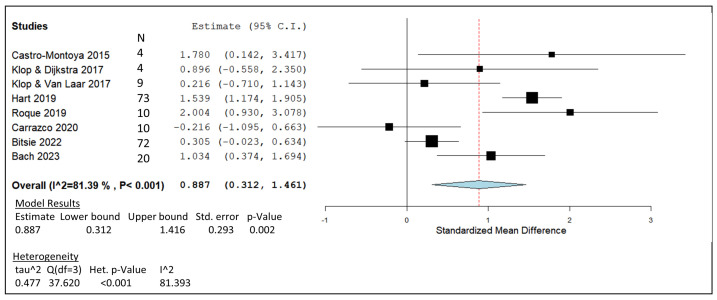
OpenMeta analysis of essential oil blends (0.04–2.5 g/kg DM) in in vivo studies over 12–96 h data collection periods. C.I.—confidence interval; std. error—standard error; Het. *p*-value—heterogeneity *p*-value. References included [86,87,88,89,90,91,106,107].

**Figure 5 animals-14-00568-f005:**
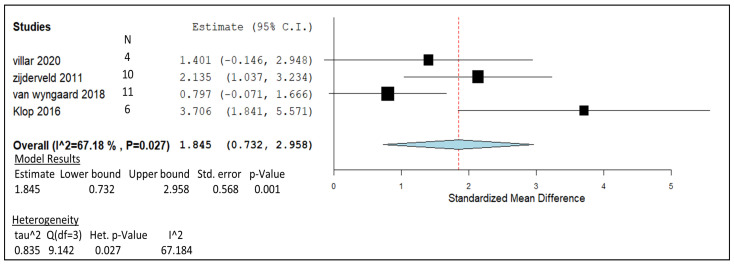
Forest plot representation of nitrate (20–23 g/kg DMI) CH_4_ gas collection meta-analysis in vivo. C.I.—confidence interval; std. error—standard error; Het. *p*-value—heterogeneity *p*-value. References included [10,48,108,109].

**Figure 6 animals-14-00568-f006:**
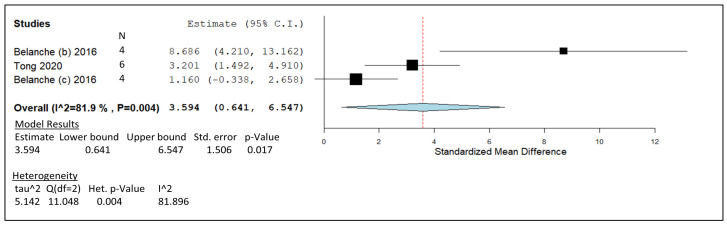
OpenMeta analysis of chitosan (16–50 mg/g DM) for CH4 mitigation after 24 h and Rusitec incubation in vitro. C.I.—confidence interval; std. error—standard error; Het. *p*-value—heterogeneity *p*-value. References included [61,110,111].

**Figure 7 animals-14-00568-f007:**
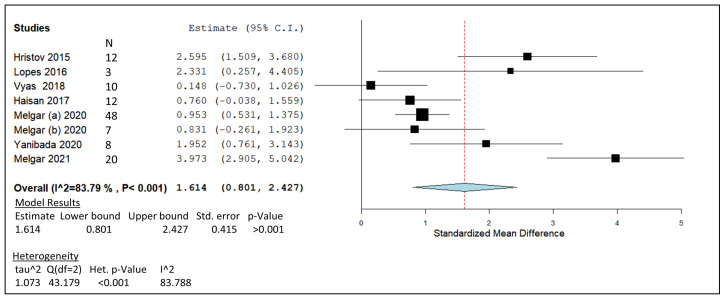
Meta-analysis investigating the 3-NOP lower-dose range of 60–75 mg/kg DMI in vivo. C.I.—confidence interval; std. error—standard error; Het. *p*-value—heterogeneity *p*-value. References included [95,96,99,100,101,102,112,113].

**Figure 8 animals-14-00568-f008:**
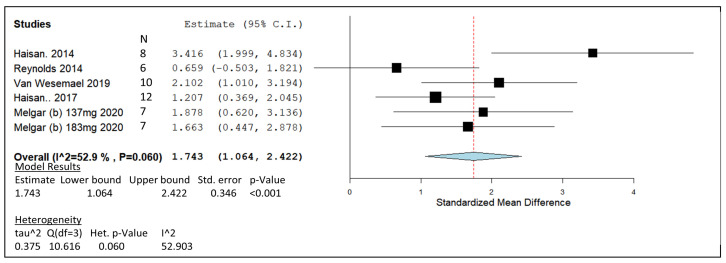
Meta-analysis investigating the 3-NOP higher-dose range of 100–183 mg/kg DMI in vivo. C.I.—confidence interval; std. error—standard error; Het. *p*-value—heterogeneity *p*-value. References included [8,96,97,98,102].

**Figure 9 animals-14-00568-f009:**
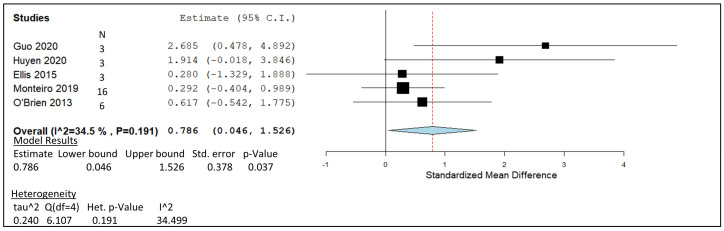
OpenMeta analysis of L. plantarum (6–9 log CFU/mL) CH_4_ production after 48–72 h of in vitro incubation. C.I.—confidence interval; std. error—standard error; Het. *p*-value—heterogeneity *p*-value. References included [71,76,120,121,122].

**Figure 10 animals-14-00568-f010:**
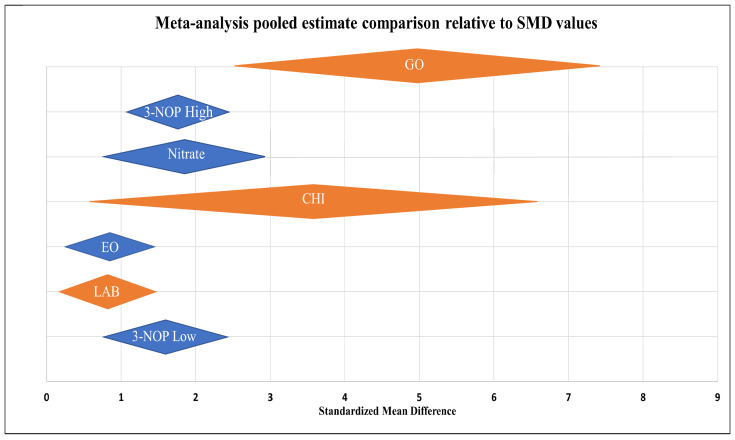
Pooled estimate of meta-analysis results relative to standardised mean difference (SMD) from in vitro studies identified by orange diamonds (
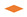
) including garlic oil (GO), L. plantarum (LAB), chi-tosan (CHI), and in vivo studies identified by blue diamonds (
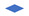
) including nitrate, essential oil blends (EOs), and 3-nitrooxypropanol at high (3-NOP High), and low doses (3-NOP Low).

**Table 1 animals-14-00568-t001:** Scopus database results based on relevant search terms and the amount of hits received within each date range.

Search Number	Scopus Search Terms	Relevant (Total) Hits	Date
1	“Methane abatement”	too broad	.
2	“Methane abatement” AND “Rumen”	7 (19)	2007–2023
3	“Methane reduction” AND “Rumen”	6 (175)	2011–2023
4	“Methane production” AND “Rumen”	19 (507)	1997–2023
5	“Methane production” AND “Rumen” AND “weight Gain”	2 (9)	2017–2018
6	“Brown seaweed” AND “Rumen”	7 (8)	2015–2019
7	“Agolin” AND “Rumen”	6 (14)	1948–2023
8	“Garlic” AND “Citrus” AND “Rumen”	4 (10)	2018–2022
9	“3-nitrooxypropanol”	9 (15)	2014–2023
10	“Ascophyllum nodosum” AND “Rumen”	7 (16)	2004–2020
11	“Laminaria digitata” AND “Rumen”	1 (8)	2018–2018
12	” Garlic oil” AND “Methane”	12 (47)	2005–2018
13	“Lactic acid bacteria” AND “Methane”	11 (19)	1988–2020
14	“Nitrate” AND “methane” AND “Rumen”	11 (149)	1972–2022
15	“Subacute rumen acidosis” AND “Rumen”	3 (240)	2007–2008
16	“Chitosan” AND “Rumen”	9 (37)	1998–2020
17	“Asparagopsis” AND “Rumen”	11 (19)	2006–2023

**Table 2 animals-14-00568-t002:** In vitro study comparison of garlic oil at concentration range of 250–300 mg/L of culture fluid (CF) in ruminal fluid from Swiss, Jersey, and Holstein cattle.

GO Study	Ruminant System	Concentration ^3^	In Vitro/In Vivo ^2^	Overall Effect ^1^	N
Soliva, 2011	Lactating brown Swiss cow	300 mg/L CF	In vitro (RUSITEC)	-GO caused almost complete inhibition of CH_4_. -Decreased protozoal numbers and increased bacterial counts.	4
Patra, 2012	Lactating Jersey cow	250 mg/L CF	In vitro	-Reduced CH_4_ emissions by around 23%. -Dose inhibitory to Archaea but did not affect feed digestibility.	3
Patra, 2015	Lactating Jersey cow	250 mg/L CF	In vitro	-Total VFA concentration decreased. -Suppressed enteric methane by 40% after 18 days.	3
Chaves, 2008	Lactating Holstein cow	250 mg/L CF	In vitro	-CH_4_ emissions from ruminal bacteria reduced 72%. -Proportion of propionate reduced.	3

^1^ GO, garlic oil; CH_4_, methane; VFA, volatile fatty acid; ^2^ RUSITEC, rumen simulation technique; ^3^ mg/L CF, miligrams per litre of culrure fluid.

**Table 3 animals-14-00568-t003:** In vivo investigation of nitrate supplementation (20–23 g/kg Dm) to Steers, Holstein–Friesian and Jersey cows.

Nitrate Study	Ruminant System	Concentration Given ^2^	In Vitro/In Vivo	Overall Effect ^1^	N
Villar, 2020	Steers	20 g/kg DM	in vivo	-Rumen protozoal concentration was reduced when including NO_3._ -Substantial decrease in CH_4_ peaks after 8 h in respiration chambers.	4
Van Zijderveld, 2011	Holstein–Friesian dairy cows	21 g/kg DM	in vivo	-Nitrate decreased methane production by 16%. -Milk protein content lowered and increased hydrogen emission.	10
Van Wyngaard, 2018	Jersey cows	23 g/kg DM	in vivo	-Milk yields decreased by 12%; concentrate DMI decreased linearly (5.5–3.7 kg/d). -CH_4_ production decreased linearly with increasing nitrate addition.	11
Klop, 2016	Holstein dairy cows	21 g/kg DM	in vivo	-Decreased enteric CH_4_ production by 23%. -Increased polyunsaturated fats and lower milk protein concentration.	6

^1^ NO_3_, nitrate; DMI, dry matter intake; CH_4_, methane; ^2^ g/kg DM, grams per kilogram dry matter.

**Table 4 animals-14-00568-t004:** Lactic acid bacteria (*L. plantarum*) in in vitro studies of Jinnan, Holstein, and Holstein–Friesian cattle ruminal fluid.

LAB Study	Ruminant System	Concentration ^2^	In Vitro/In Vivo	Overall Effect ^1^	N
Guo, 2020	Jinnan cattle	1.0 × 10^6^ cfu/g (fresh weight)	in vitro	-Lowered the ratio of CH_4_ output to total VFAs. -Increased acetate and propionate, total VFA, DM-D, and NDF-D as compared with that of the control after 72 h in vitro incubation.	3
Huyen, 2020	Lactating Holstein–Friesian cow	1.0 × 10^6^ cfu/g (fresh weight)	in vitro	-CH_4_ production was lower for LAB when used as silage inoculants, compared to being used as directly fed microbials. -Increased the in vitro DM and organic matter (OM) degradability both in the fresh ration and rain treated ration.	3
Ellis, 2016	Holstein–Friesian	‘1.0 × 10^6^ cfu/mL (In 60 mL of buffered rumen fluid)	in vitro	-No significant effect of LAB treatment on OM digestibility, cumulative gas or CH_4_ production.	3
Monteiro, 2020	Lactating Holstein cow	1.35 × 10^9^ cfu/g DM	in vitro	-No significant changes on CH_4_ production. -Lower CO_2_ production linked with total VFA reduction over 24 and 48 h.	16
O’Brien, 2013	Holstein cow	‘1.0 × 10^8.3^ cfu/mL (In 100 mL glucose-yeast medium)	in vitro	-Molar proportions of propionic acid increase and lower levels of acetic and butyric acid. -Decrease in total VFA concentration (17%) (mM) and CH_4_ output (68%) (ml 24 h^−1^).	6

^1^ CH4, methane; VFA, volatile fatty acid; DM-D, dry matter digestibility; NDF-D, neutral detergent fibre digestibility; LAB, lactic acid bacteria; DM, dry matter; OM, organic matter; CO_2_, carbon dioxide; ^2^ cfu/mL, colony-forming units per millimetre.

**Table 5 animals-14-00568-t005:** Chitosan (16–50 mg/g DMI inclusion rate) relevant studies including Holstein and Holstein-Friesian cattle rumen fluid investigated in vitro.

Chitosan Study	Ruminant System	Concentration Given ^2^	In Vitro/In Vivo	Overall Effect ^1^	N
Belanche, 2016b	Holstein–Friesian	50 mg/g DM	In vitro (RUSITEC)	-Decreased rumen methanogenesis by 42%. -Promoted shift in fermentation pattern towards propionate production.	4
Tong, 2020	Lactating Holstein cow	16 mg/g DM	In vitro	-Propionate concentration was significantly increased, and acetate proportion was decreased. -CH_4_ reduced by replacing fibrolytic bacteria with amylolytic bacteria.	6
Belanche, 2016c	Holstein–Friesian	50 mg/g DM	In vitro	-Fermentation shifted towards propionate production. -Lower CH_4_ (23%) and protozoal activity (56%).	4

^1^ CH_4_, methane; ^2^ mg/g DM, milligrams per gram dry matter.

**Table 6 animals-14-00568-t006:** Essential oil blends (0.04–2.5 g/kg DM) in vivo publications with feed supplementation to dairy cattle. DM—dry matter; CP—crude protein; ADF—acid–detergent fibre.

AR/MO Ruminant Study	Ruminant System	Concentration Given ^2^	In Vitro/In Vivo	Overall Effect ^1^	N
Klop, Vaan Laar-Van Shuppen, 2017b	Lactating Holstein cow	0.05 g/kg DM	in vivo	-Average CH_4_ production was decreased by 8% when AR was supplemented to associated doner animals for 3 weeks. -No negative effects on dietary mass intake for cows receiving AR diet.	9
Klop, Dijkstra, 2017a	Lactating dairy cows	0.17 g/kg DM	In vivo	-CH_4_ production lowered after first period of 2 weeks only. -Higher proportions of acetate and propionate with AR supplementation.	4
Hart, 2019	Holstein–Friesian cows	0.05 g/kg DM	In vivo	-Yields of milk fat, protein, lactose, and solids were higher for AR-fed cows. -CH_4_ output was reduced by 27 g/day with AR compared to control treatment.	73
Castro Montoya, 2015	Lactating Holstein cow	0.05 g/kg DM	In vivo	-Milk production displayed a linear decrease towards the end of the study. -Addition of AR accounted for 15% (g/d) decrease in CH_4_ over the experimental period.	4
Carrazco, 2020	Lactating Holstein cow	0.04 g/kg DM	In vivo	-CH_4_ yield was not significantly reduced by essential oil blend. -Ruminant production parameters did not differ with AR supplementation.	10
Bach, 2023	Lactating Holstein cow	0.04 g/kg DM	In vivo	-AR supplementation decreased CH_4_ yield (L/kg DM) by 12.3%. -Feed efficiency (ECM/DMI) and diversity of microbiome was lower with AR supplemented cows.	20
Roque, 2019	Angus x Hereford steers	1.6 g/kg DM	In vivo	-CH_4_ yield decreased by 13.3% with MO supplementation. -DMI, ADG, and feed efficiency remained unchanged with essential oil blend supplementation.	10
Bitsie, 2022	Angus x Simmental steers	2.5 g/kg DM	In vivo	-CH_4_ yield (g/kg DMI) decreased by 24.6% with MO. -MO increased DM, CP, and ADF apparent digestibility.	72

^1^ CH_4_, methane; AR, Agolin ruminant; L/kg DM, litres per kilogram dry matter; ECM, energy corrected milk; DMI, dry matter intake; MO, Mootral; ADG, average daily gain; DM, dry matter; CP, crude protein; ADF, acid–detergent fibre; ^2^ g/kg DM, grams per kilogram dry matter.

**Table 7 animals-14-00568-t007:** 3-Nitrooxypropanol low dose (60–75 mg/kg DM) study search results including Holstein cattle and steers in vivo.

3-NOP Low-Dose Study	Ruminant System	Concentration Given ^2^	In Vitro/In Vivo	Overall Effect ^1^	N
Hristov, 2015	Lactating Holstein cow	60 mg/kg DM	in vivo	-Milk protein and lactose increased by 3-NOP. -Increased body weight. -CH_4_ emissions decreased by 30% lower than control.	12
Lopes, 2016	Lactating Holstein cow	60 mg/kg DM	In vivo	-Acetate to propionate ratio was lower when treated with 3-NOP. -Proportions of methanogens decreased. -Inhibition of enteric methane with increased hydrogen emissions.	3
Vyas, 2018	Crossbred steers	75 mg/kg DM	In vivo	-Lowered total CH_4_ emissions with increased 3-NOP supplementation. -No negative effects on DMI.	10
Haisan, 2017	Lactating Holstein cow	68 mg/kg DM	In vivo	-Molar proportions of acetate to propionate reduced in dose dependant manner. -CH_4_ yield (g/kg DM) decreased by 23%.	12
Melgar, 2020a	Lactating Holstein cow	60 mg/kg DM	In vivo	-3-NOP decreased CH_4_ yield by 21% over a 15-week treatment period. -Hydrogen emissions were increase by 48-fold.	48
Melgar, 2020b	Lactating Holstein cow	73 mg/kg DM	In vivo	-Hydrogen emissions increase over 7-fold with 3-NOP inclusion. -CH_4_ yield decreased by 16% g/kg DM.	7
Yanibada, 2020	Lactating Holstein cow	60 mg/kg DM	In vivo	-CH_4_ yield mitigated by 21.7% over 5-week treatment. -No changes in milk yield or composition.	8
Melgar, 2021	Lactating Holstein cow	60 mg/kg DM	In vivo	-3-NOP decreased CH_4_ yield by 27% compared to the control. -Hydrogen emissions were increased 6-fold with 3-NOP inclusion. -Increased milk fat yield with treatment.	20

^1^ 3-NOP, 3-nitrooxypropanol; CH_4_, methane; DMI, dry matter intake; g/kg DM, grams per kilogram dry matter; ^2^ mg/kg DM, milligrams per kilogram dry matter.

**Table 8 animals-14-00568-t008:** 3-Nitrooxypropanol high dose (100–183 mg/kg DM) relevant papers and results on Angus, Holstein-Friesian and Holstein cattle in vivo.

3-NOP High Dose Study	Ruminant System	Concentration Given ^2^	In Vitro/In Vivo	Overall Effect ^1^	N
Haisan 2014	Holstein lactating	126.9 mg/kg DM	In vivo	-CH_4_ production was reduced by 10.62 g/kg DMI. -Cattle fed 3-NOP gained more body weight. -Reduction in acetate to propionate ratio was observed.	11
Reynolds 2014	Holstein-Friesian cows	135.1 mg/kg DM	In vivo	-Decrease in acetate to propionate ratio. -Dry matter, organic matter, acid detergent fibre and energy digestibility were reduced.	4
Van Wesemael 2019	Holstein-Friesian lactating	100 mg/kg DM	In vivo	-CH_4_ production was 28% lower for basal diet treated with 3-NOP and 23% lower for concentrate diets.	6
Haisan 2017	Holstein lactating	132 mg/kg DM	In vivo	-CH_4_ yield (g/kg DM) was decreased by 36.7% with high 3-NOP. -Apparent total-tract digestibility significantly increased with 3-NOP.	12
Melgar 2020b	Holsten lactating	137 mg/kg DM + 183 mg/kg DM	In vivo	-CH_4_ yield decreased by 36% and 31.8% with 137 and 183 mg/kg DM 3-NOP, respectively. -Hydrogen emission production increased 9-fold and 7-fold with 137 and 183 mg/kg DM 3-NOP, respectively. -Milk fat % increased with 3-NOP supplementation.	7

^1^ CH_4_, methane; g/kg DMI, grams per kilogram dry matter intake; 3-NOP, 3-nitrooxypropanol; g/kg DM, grams per kilogram dry matter; ^2^ mg/kg DM, milligrams per kilogram dry matter.

## Data Availability

Data are contained within the article.

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
