# Peer review of "A Review of Potential Feed Additives Intended for Carbon Footprint Reduction through Methane Abatement in Dairy Cattle"

_animals, 2024, doi:10.3390/ani14040568_

Round 1

Reviewer 1 Report

Comments and Suggestions for Authors

The topic of this meta-analysis review is an interesting idea because it focuses comparison between different feed additives' effects on GHG, especially methane abatement in Dairy Cattle.

General comments:

- I suggest trying to improve the abstract and simple abstract to clear the topic sentence and the aim of the study.

- I suggest being accurate especially when selecting the types of additives. The authors grouped the feed additives based on in vitro or in vivo available studies and conclusions were determined based on their effectiveness in live subjects or their potential efficacy in live animal trials.

My suggestion: I would like to suggest adding another difference in comparing the feed additives, the adaptation of the feed additives. Especially, garlic oil (GO) many studies reported that some microorganisms inside the rumen adapted and resistant to this oil.   

Specific comments:

- the methods used in this study were valid and reliable to use a good number of feed additives 7 rumen additives like garlic oil (GO), nitrate, Ascophyllum nodosum (AN), Asparagopsis (ASP), Lactobacillus plantarum (LAB), chitosan (CHI), Essential oils (EO) and 3-nitrooxypropanol (3-NOP).

My comment: I suggest redesigning and grouping the feed additives into 4 categories

1-      Essential oils (EO),

2-      Chemical compounds (nitrate, 3-nitrooxypropanol and chitosan (CHI)),

3-      Algae or seaweed (Ascophyllum nodosum (AN), Asparagopsis (ASP),

4-      Bacteria (Lactobacillus plantarum (LAB)).

- Line 159: I suggested that: the authors support the introduction by describing the chemical structure of the different feed additives used in this review.

- Line 163 authors describe the mechanism of action of garlic oil and its compounds are primarily related to direct inhibition of methanogenesis in methanogenic Archaea.

My comment: there is no data about the adaptation of garlic oil (GO) but many studies reported that methanogenic Archaea inside the rumen adapted and resistant to garlic oil after one month so, recommended adding different EO or blend of these oils to reduce methane emission.   

The discussion needs more explanation of your findings. I suggest a design form to explain and conclude the mechanism of the Garlic oil and EO supplementation on methane production.

- The conclusions answer the aims of the study. Also, the authors conclude the findings and recommend the best level of feed additives that can be added to in vitro and in vivo trials.

Reviewer 2 Report

Comments and Suggestions for Authors

The importance of reducing the production of greenhouse gases, including in agriculture, is an extremely important environmental task. Cattle are one of the most important sources of greenhouse gases, so minimizing methane emissions from cows is an important goal. Various strategies are used to achieve this, the most important of which is the use of feed additives that reduce methane production by cows. Authors chosen seven rumen additives for an enteric methane mitigating comparison study including garlic oil (GO), nitrate, Ascophyllum nodosum (AN), Asparagopsis (ASP), Lactobacillus plantarum (LAB), chitosan (CHI), Essential oils (EO) and 3 -nitrooxypropanol (3-NOP) and chose the leading system, which is of obvious interest for further research. The article is written logically, clearly, in good language and contains links, but I suggest that the authors strengthen the links a little. The article is well illustrated, the experiment was carried out methodically correctly, and statistics were used where necessary. I am sure that this article will be of interest to many experts and will be well cited. I suggest publishing this article

Reviewer 3 Report

Comments and Suggestions for Authors

The manuscript is well structured and easy to read, despite being extensive. There are some inaccuracies that were detected that require corrections.

Manuscript Title:

"A Review of Potential Feed Additives Intended for Carbon Footprint Reduction Through Methane Abatement in Dairy Cattle."

Title Feedback:

The title does not reflect the work. Currently, there are many additives used and researched in methane reduction. This work only reflects the potential of 8 in different areas. They should therefore adapt the title to the work presented.

I suggest revising the manuscript's title.

Simple summary

The summary should be concrete and objective.

They should immediately identify the objective of the work and the additives used.

Abstract:

The abstract is well written and summarizes the entire work. But there is a lack of agreement between the number of additives studied, while here they mention 7 additives (L27) in the rest of the manuscript there are always 8 additives mentioned.

Introduction:

In the introduction there are only two notes that, in my opinion, are necessary to add:

1 – Justify the reason why you selected the 8 additives

2 – Write clearly at the end of the introduction the objective of the work.

Materials and Methods:

They must indicate which are the large ruminants: Cattle, Buffaloes, etc.. After reading the work, I notice that only beef and dairy cattle are mentioned, why don't they just use this expression and refer to large ruminants? is that it misleads.

The tables are poorly formatted, the lines do not match, you must review all table formatting.

Results:

Below the tables must contain the captions of the acronyms used in each table.

They must review the formatting of all tables

Please note that the subtopic numbers are not correct. It goes from 3.1 to 1.1 and then to 3.2. Correct this inaccuracy.

Discussion:

 The discussion is well done, they should approach this discussion using more recent publications.

Conclusion:

Simple conclusion, but with a good summary of the entire discussion

References:

Despite being a review work, 140 bibliographic references are a somewhat high numberFinal Note:

It is a well-prepared work, there are just some inaccuracies that must be improved before it can be published later.

The biggest limitation of this work is that in a review work only around 24% of the references are less than 5 years old. They must review this situation in order to increase this ratio to a minimum of 30 -35%, as it is necessary in these review works to know the past, but above all to make known what has been done more recently, as this topic has been widely discussed in recent years. 
